# Total Pancreatectomy with Autologous Islet Cell Transplantation—The Current Indications

**DOI:** 10.3390/jcm10122723

**Published:** 2021-06-20

**Authors:** Beata Jabłońska, Sławomir Mrowiec

**Affiliations:** Department of Digestive Tract Surgery, Medical University of Silesia, 40-752 Katowice, Poland; mrowasm@poczta.onet.pl

**Keywords:** total pancreatectomy, islet transplantation, autotransplantation

## Abstract

Total pancreatectomy is a major complex surgical procedure involving removal of the whole pancreatic parenchyma and duodenum. It leads to lifelong pancreatic exocrine and endocrine insufficiency. The control of surgery-induced diabetes (type 3) requires insulin therapy. Total pancreatectomy with autologous islet transplantation (TPAIT) is performed in order to prevent postoperative diabetes and its serious complications. It is very important whether it is safe and beneficial for patients in terms of postoperative morbidity and mortality, and long-term results including quality of life. Small duct painful chronic pancreatitis (CP) is a primary indication for TPAIT, but currently the indications for this procedure have been extended. They also include hereditary/genetic pancreatitis (HGP), as well as less frequent indications such as benign/borderline pancreatic tumors (intraductal papillary neoplasms, neuroendocrine neoplasms) and “high-risk pancreatic stump”. The use of TPAIT in malignant pancreatic and peripancreatic neoplasms has been reported in the worldwide literature but currently is not a standard but rather a controversial management in these patients. In this review, history, technique, indications, and contraindications, as well as short-term and long-term results of TPAIT, including pediatric patients, are described.

## 1. Introduction

Total pancreatectomy is a major complex surgical procedure involving removal of the whole pancreatic parenchyma and duodenum. It leads to lifelong pancreatic exocrine and endocrine insufficiency. Currently, it is performed for both benign and malignant pancreatic diseases. In the treatment of pancreatic exocrine insufficiency (PEI), oral supplementation of pancreatic enzymes for the reduction of steatorrhea is used. The control of surgery-induced diabetes (type 3) requires insulin therapy. The correct diabetes control is very important for patients, because not controlled or poorly controlled diabetes leads to numerous serious complications such as an ischemic heart disease, brain stroke, peripheral artery disease (PAD), renal insufficiency, blindness, secondary to diabetic microangiopathy, macroangiopathy, and neuropathy. Therefore, total pancreatectomy with autologous islet transplantation (TPAIT) is performed in order to prevent postoperative diabetes and its serious complications. It is very important whether it is safe and beneficial for patients in terms of postoperative morbidity and mortality, and long-term results including quality of life (QoL) [1].

The first reports regarding TPAIT, initially in animals, and subsequently in humans, were described in the 1970s of the previous century. Since then, numerous reports, initially case reports, subsequently retrospective and prospective observational and control studies, meta-analyses, and randomized prospective trials have been published. The aim of this paper is to review the most important publications regarding TPAIT.

## 2. Methods of the Literature Searching

We reviewed a PubMed database for the following search terms and meSH terms from 1975 to 2021: total pancreatectomy with/and pancreatic islet transplantation, pancreatic islet autotransplantation, chronic pancreatitis, hereditary genetic pancreatitis, intraductal papillary mucinous neoplasm. We selected English, full-text publications which exactly corresponded to the subject of our interest. We noted that most of the studies were performed on dogs. Subsequently, first, case reports were described, followed by case series, retrospective observational, control, and prospective studies. Initially, singular articles related to TPAIT were published, and the number of publications has increased gradually.

## 3. History of Total Pancreatectomy with Autologous Pancreatic Islet Transplantation

The first successful case of intrasplenic TPAIT performed on canine models was described by Mirkowith et al. in 1976 [2]. In this experimental study, a dispersed pancreatic tissue was prepared by collagenase digestion without separation of exocrine and endocrine pancreatic components. This unpurified pancreatic tissue was infused into the dogs’ spleen. In 20 out of 25 dogs, a normalized serum glucose was reported [2]. The first report of human TPAIT was published by Najarian, Sutherland, and co-authors from the University of Minnesota in 1980 [3]. The authors applied the technique of TPAIT, previously used in dogs, to ten patients with chronic pancreatitis (CP) with small pancreatic ducts and severe, intractable pain. Subtotal pancreatectomy (>95% of the pancreas volume) was performed, and the removed pancreas was minced, dispersed by collagenase digestion, and infused into the portal vein <2 1/2 h after pancreatic resection. Pain relief was achieved in all patients. One patient died of a complication not related to this procedure. In the remaining nine patients, insulin independence or the significant decrease of insulin requirements was noted. The proper function of transplanted pancreatic islets was confirmed by C-peptide investigation [3]. In the first years, most publications on TPAIT regarded studies carried out on animal models (dogs, pigs, rats). Most studies were performed on dogs. The pancreatic islets were autotransplanted either directly into the splenic pulp, or into the splenic artery or the portal vein [4,5,6,7,8,9,10,11]. In 1978, based on studies performed on canine models, Kretschmer et al. [5] concluded that a direct implantation of pancreatic islets into the splenic pulp was better than transplantation into the portal vein or splenic artery because the splenic circulation was not disturbed, and portal hypertension was avoided, as well as a postoperative glucose control was better. The next reports on human TPAIT were published in the 1980s of the 20th century [12,13]. Initially, the studies involved patients operated on for intractable pain due to CP. In 1990, Farney et al. [14] described results of the study including 26 patients who underwent autotransplantation of dispersed pancreatic islet tissue combined with total or near-total pancreatectomy for treatment of CP between 1977 and 1991. Pancreatic islets were injected intraportally in 22 patients and into the renal subcapsule in two patients. In the next years, the pioneer surgical center at the University of Minnesota was evaluated. In 1995, the authors published results of islet autotransplantation performed in 48 patients following total/near-total (>95%) pancreatectomy (*n* = 43) and partial pancreatectomy (*n* = 5) between 1977 and 1995. The authors showed that islet autotransplantation could prevent long-term diabetes in more than 33% of patients [15]. Moreover, in 1995, a case of pregnancy following TPAIT was reported. This case report confirmed a high QoL in patients following TPAIT [16]. In 1996, a case of a 12-year-old boy who underwent successful TPAIT for intractable pain caused by idiopathic CP was reported. Postoperatively, complete relief of abdominal pain was achieved. The patient remained insulin-independent, with normal fasting blood glucose and hemoglobin A1C levels, for 2 1/2 years [17]. In 1997, a case of transcontinental shipping of pancreatic islets for autotransplantation after total pancreatectomy was reported. Pancreatectomy is performed at most hospitals, but islets are prepared at only a few centers. Therefore, the possibility of successful transcontinental shipping of pancreatic islets for autotransplantation after total pancreatectomy is very important in order to spread this procedure [18]. The first islet autotransplant program in the United Kingdom (the Leicester experience) and the first series in the world to use the spleen as a site for the islet graft was reported in 1999. Over an 11-month period, seven patients underwent TPAIT for CP. In six patients, islets were embolized into the liver via the portal vein (median transplanted volume = 8.5 mL). Three patients received islets into the splenic sinusoids via a short gastric vein. One patient received islets into the spleen alone [19]. In 2001, good long-term results of intrahepatic autoislet transplantation in six CP patients with stable beta-cell function and normal levels of blood glucose and HbA1C for up to 13 years were presented [20]. In 2003, the authors from the Leicester general hospital described the results of TPAIT performed in 40 patients with CP. The follow-up times ranged from 6 months to 7 years [21]. In 2008, the long-term results of 46 patients were published [22]. Moreover, in 2003, the authors from the University of Cincinnati College of Medicine published the results (22 patients) [23]. In 2004, a case report of TPAIT with the use of a temporarily exteriorized omental vein was reported by the authors from the University of Minnesota [24]. The indications for TPAIT have been extended. In 2012, an emergency autologous islet transplant after a traumatic Whipple operation and subsequent total pancreatectomy performed for a 21-year-old patient who was wounded with multiple abdominal gunshot wounds was reported [25]. In 2013, Balzano et al. described extended indications for islet autotransplantation to grade C pancreatic fistula (treated with completion or left pancreatectomy, as indicated), total pancreatectomy as an alternative to high-risk anastomosis during pancreaticoduodenectomy, and distal pancreatectomy for benign/borderline neoplasm of pancreatic body-neck [26]. Moreover, the procedure approach has been modified through the years. In 2012, the first robot-assisted pancreatoduodenectomy with preservation of the vascular supply for autologous islet cell isolation and transplantation was reported. So far, both open and robotic-assisted TPAIT are performed [27].

## 4. Indications and Contraindications for TPAIT

In 2013, the participants of PancreasFest [28] published recommendations for TPAIT. According to the authors, severe CP and recurrent acute pancreatitis (RAP) with intractable pain which leads to low QoL in these patients are indications for TPAIT. The authors recommend TPAIT in patients for whom previous conservative, endoscopic, and surgical treatment was not effective. Therefore, patients with small duct CP are main candidates for this treatment method. In these patients, the main goal of TPAIT is to relieve the pain and to improve QoL (by total pancreatectomy) as well as to prevent postoperative type 3 diabetes (by autologous islet transplantation) [28]. In the authors’ opinion, severity, frequency, and duration of pain, requirement of narcotic drugs, decreased QoL, residual pancreatic islet function, rate of disease progression, and patient’s age are important in the choice of optimal timing of the procedure [28]. It should be pointed that in CP, endoscopic and surgical treatments should be taken into consideration. Endoscopic treatment (ET) is the most useful in patients with pancreatic duct lithiasis, obstruction, and dilation. It should be the first-line option because it is less invasive than surgery. Surgery such as drainage operations (Puestow’s, Partington-Rochelle’s, Duval’s procedures), resectional operations (partial and subtotal or total pancreatectomies), resections with extended drainage (Beger’s, Frey’s procedures) should be the first-line option in patients for whom ET has failed or in those with a pancreatic mass with suspicion of malignancy. In general, TPAIT should be considered in patients who have failed other operations or in patients with small duct or minimal change disease [29].

Hereditary/genetic pancreatitis (HGP) is a specific CP kind that is an indication for TPAIT. It is known that in patients with hereditary pancreatitis or *PRSS1* gene mutations an increased risk of pancreatic cancer is noted. The goal of TPAIT in these patients is to remove a pancreas and prevent pancreatic cancer [28].

HGP is caused by a mutation in the gene encoding cationic trypsinogen (protease serine 1, or *PRSS1*), mapped to the 7q35 region of the long arm of human chromosome 7. Pathogenic mutations in the *PRSS1* are associated with more than an 80% chance for recurrent acute and/or chronic pancreatitis, as well as an exceptionally high lifetime risk (estimated at 40%) for pancreatic ductal adenocarcinoma. Because of a high cancer risk, these patients are distinct from the general TPAIT population. TPAIT in these patients is considered typically in young patients with first symptoms (often <10 years) and a higher lifetime risk for pancreatic cancer. Patients with *PRSS1*-mediated CP are increasingly being considered for TPIAT when pain and disability are severe. So, in HGP patients, the goal of TPAIT is not only to relieve a pain and increase QoL, but also to decrease a pancreatic cancer risk. A strong association between HGP duration and the diabetes outcomes has been observed. In a comparison of HGP patients with those patients with no known genetic risk factors, a smaller and more fibrotic pancreas, lower islet mass, and lower probability of ever achieving insulin independence was noted in HGP patients compared to non-genetic CP [30]. A study by Chinnakotla et al. [31] including 80 HGP patients showed that TPAIT in HGP patients provides long-term pain relief (90%) and preservation of β-cell function. HGP patients with a high lifetime risk of pancreatic cancer should be considered earlier for TP-IAT before pancreatic inflammation results in a higher degree of pancreatic fibrosis and islet cell function loss. Therefore, these patients are typically younger compared to the patients undergoing TPAIT for other indications [31]. A study by Bellin et al. including 64 HGP patients showed that the postoperative outcome was adversely impacted by older age and prolonged disease. In particular, islet mass was lower and risk of diabetes high in older patients with a prolonged disease. This should be considered during counseling of this subgroup of TPIAT recipients for surgical management [30].

Although small duct painful CP is the common indication for TPAIT, some other less frequent indications for this procedure have been described such as total pancreatectomy as treatment for severe pancreatic fistulas, extensive distal pancreatectomy for neoplasms of the pancreatic isthmus, or total pancreatectomy because of a preoperative assessment of the increased anastomotic risk (for example, fatty or soft pancreas, patient’s comorbidities). In 2016, Balzano et al. [32] presented criteria of extended indications for TPAIT included in the Milan Protocol. This protocol involves the following clinical indication groups: (1) CP and RAP; (2) so-called “high-risk pancreatic stump” in patients undergoing pancreaticoduodenectomy in whom pancreatic anastomosis is associated with a high risk of dehiscence (according to the assessment of the chief surgeon), based on the presence of a narrow main pancreatic duct and soft and/or frail pancreatic texture; (3) extensive distal pancreatectomy for benign/borderline neoplasm located at the pancreatic isthmus (as an alternative procedure to central pancreatectomy); (4) severe complications following partial pancreatectomy, such as Grade C postoperative pancreatic fistula (POPF), according to the definition of the International Study Group on Pancreatic Fistula, requiring relaparotomy with completion of pancreatectomy or distal pancreatectomy [32].

According to PancreasFest [28], TPAIT is contraindicated in patients with C-peptide negative diabetes, type 1 diabetes, portal vein thrombosis, portal hypertension, advanced liver disease or cardiopulmonary disease, and pancreatic cancer. Besides the above-mentioned contraindications, participants of PancreasFest presented psychosocial contraindications for TPAIT such as active alcohol abuse, active illicit substance use, and untreated/uncontrolled psychiatric disease that could impair the patient’s ability to adhere to complicated medical management (reduction in the doses of analgetic drugs, pancreatic enzyme therapy, diabetes control, and frequent follow-up) [28].

According to the Milan Protocol, pancreatic malignancy is a contraindication for TPAIT. The following contraindications are described: (1) presence of any multifocal pancreatic neoplasm at preoperative imaging or intraoperative assessment, (2) diagnosis of intraductal papillary mucinous neoplasm (IPMN), unless the absence of multifocal lesion is demonstrated by endoscopic ultrasound, (3) pathologic pancreatic resection margin, including any degree of dysplasia or ductal disepithelialization, (4) diagnosis of multiple endocrine neoplasm, (5) any medical condition that, in the opinion of the surgeon, might have a negative influence on procedure safety [32].

Clinical contraindications for pancreatectomy with autologous islet transplantation in patients with pancreatic neoplasms are controversial. There are reports showing the successful use of this procedure in patients with pancreatic neoplasms (including IPMN, insulinoma, pancreatic and ampullary cancer, renal cell carcinoma (RCC) metastasis to pancreas) in the literature [33,34,35,36,37,38,39,40,41,42,43,44,45,46,47,48,49,50,51,52]. Ris et al. [34] described 25 patients undergoing extended pancreatectomy and islet autotransplantation for benign pancreatic tumors. There were 14 patients with benign tumors located in the pancreatic isthmus including 10 (4 mucinous and 6 serous) cystadenomas and 4 neuroendocrine tumors (3 insulinomas and 1 non-secreting neuroendocrine tumor) in this cohort. Malignancy was ruled out in each tumor, both based on the intraoperative and postoperative histological findings. The actuarial 10-year survival rate was 100% [16]. Oberholzer et al. [35] described the first cases of extended left pancreatectomy with autologous islet transplantation. The authors considered that the risk of tumor dissemination by this procedure was low because of the presence of a single tumor, intraoperative frozen sections assuring tumor-free resection margins, as well as no invasion of surrounding tissue, but there was no information regarding long-term results in these patients [35]. Zureikat et al. [38] described ten robotic-assisted TPAIT in patients with indications including IPMN (*n =* 6) and pancreatic adenocarcinoma (*n =* 1). In this study, only short-term results were reported [38]. Balzano et al. [32] analyzed results of TPAIT performed for extended indications than in the Milan Protocol, including treatment for severe pancreatic fistulas (*n* = 21), extensive distal pancreatectomy for neoplasms of the pancreatic neck (*n* = 19) or pancreaticoduodenectomy because of the high risk of pancreatic fistula (*n* = 32). In this study, in 31 of 58 patients, malignant pancreatic or periampullary neoplasms were reported, and ex novo liver metastases following surgery were noted in only 3 patients (median follow-up 914 ± 382 days) [32]. Relapse was observed only in 5 (12.9%) of 31 patients who were disease-free after surgery. In a comparison of overall survival and disease-/progression-free survival of patients with ductal carcinoma treated with islet autotransplantation to those of patients with ductal carcinoma who had surgery in the same period of time but did not receive islet autotransplantation, patients undergoing islet autotransplantation had a better survival than did patients without islet autotransplantation at a similar stage of disease. In fact, malignancy has been considered an absolute contraindication for islet autotransplantation because of the risk of disseminating cancer cells through the infusion of islets, which may still contain some exocrine cells even after purification. Therefore, prospective randomized trials are needed to assess the long-term oncological results of TPAIT [32]. There are a few reports on the use of islet autotransplantation in patients with a malignant neoplasm such as ampullary adenocarcinoma (T3N1M0) with postoperative complications (1 year of follow-up with no cancer recurrence) [40], ampullary adenocarcinoma pT1N0MX G1 with postoperative complications (3 months of follow-up with no cancer recurrence) [41], pancreatic cancer with postoperative complications (1 year of follow-up with no cancer recurrence) [42], and pancreatic cancer (T3pN1G2) with postoperative complications (died due to cancer recurrence 2 1/2 years following surgery) [43]. Kocik et al. [46] presented five patients undergoing completion pancreatectomy with autologous islet transplantation as treatment of POPF following pancreaticoduodenectomy due to malignant neoplasms including pancreatic adenocarcinoma, cholangiocarcinoma, and papilla of Vater cancer. Two of five patients were still alive 9 and 28 months after surgery. Savari et al. [50] presented a case of TPAIT in a 54-year-old male with an invasive 2-cm ampullary adenocarcinoma (T3N0M0) and splenic vein thrombosis, pseudocyst, and abscess in the pancreatic tail. Six months after surgery, disseminated metastatic disease (adenocarcinoma of the skin in the scar at the site of the drain placement in the left upper abdomen and pulmonary metastases) was noted. Gala-Lopez et al. [51] reported a case of TPAIT in a 70-year-old female with multifocal pancreatic metastases from RCC. In this patient, insulin independence and no tumor recurrence were noted one year following surgery. Renaud et al. [47] published results of the experimental study including patients requiring pancreatectomy for pancreatic ductal adenocarcinoma (PDAC). Immunocompromised mice were transplanted with pancreatic cells isolated from the nonmalignant part of the surgical specimen (experimental group). Results were compared with pancreatic tumor implants (control group). Pancreatic grafts were explanted at 6 weeks for histological analyses. This study included 9 patients, and 31 mice were transplanted. In the experimental group, explants were microscopically devoid of the tumor cell, and no metastasis was observed. In the control group, all explants were composed of the tumor. In this study, the authors demonstrated the absence of local and distant spreading of malignant cells after pancreatic islets xenograft was isolated from PDAC patients. In the authors’ opinion, these results confirmed the oncological safety of TPAIT as a valuable alternative to partial pancreatectomy for PDAC patients with a high risk of POPF [47]. Despite these above-mentioned findings, the oncological safety of TPAIT for PDAC patients should be assessed by prospective randomized trials in order to recommend this procedure routinely.

Sakata et al. [44] presented TPAIT as a treatment of pancreatic arteriovenous malformation (AVM) performed in three males in ages of 39, 52, and 59 years, achieving good results in two of them. Wang et al. [45] reported a case of successful TPAIT performed on a 16-year-old boy with intractable pain due to CP coexisting with ulcerative colitis and primary sclerosing cholangitis (PSC). Desai et al. [48] described the first report of successful TPAIT for pancreatic cystosis on a 21-year-old woman with pancreatic cystic fibrosis, with insulin independence 11 months following surgery. St Onge et al. [49] presented TPAIT on a 4-year-old female with pancreatic cystic fibrosis, with insulin independence one year following surgery.

In conclusion, small duct CP with intractable pain is a common indication for TPAIT. Recently, primary indications for TPAIT have been extended and include selected patients with benign pancreatic tumors and other rare pancreatic diseases (such as pancreatic AVM and cystic fibrosis). Currently, some primary contraindications (such as IPMN, endocrine tumors) became indications for TPAIT. The use of TPAIT in malignant pancreatic and peripancreatic neoplasms is controversial, and currently it is not a standard management. Large, randomized trials are needed to assess long-term oncological results (such as disease-free survival).

## 5. Pancreatic Islet Isolation and Transplantation—Technical Aspects

The standard procedure preceding islet transplantation involves total or partial pancreatectomy (in two steps or en-block, with or without splenectomy). Open laparotomy is the most frequent approach for this procedure, although laparoscopic and robotic-assisted laparoscopic approaches have been described [52,53,54,55,56,57,58]. The gastroduodenal artery and the splenic artery and vein are preserved until just prior to resection in order to minimize warm ischemia to the islet tissue. The spleen may be preserved using the short gastric vessels, although sometimes anatomic conditions allow pancreatectomy without damage to the splenic vessels [52]. After pancreatic devascularization, the pancreas is promptly removed and placed in cold preservation solution. The duodenum and spleen are separated from the pancreas. The pancreatic duct is assessed for integrity, cannulated, and flushed with cold preservation solution, and blood is flushed from the major vessels. Then, the organ is packed in cold preservation solution and transported to the islet isolation laboratory [52,59]. The goal of islet isolation is to digest the pancreas and disrupt the exocrine pancreatic tissue in order to release relatively pure islets into a small tissue volume that can be safely infused into the portal vein [52,60]. The pancreas is first distended by intraductal injection of collagenase, followed by gentle mechanical dispersion using the semi-automated Ricordi method, freeing islets from the exocrine tissue [52,61,62]. The Ricordi method was introduced in 1989 and meets the following criteria: minimal trauma of the islets, continuous digestion in which the islets that are progressively liberated can be saved from further enzymatic action, minimal human intervention in the digestion process, and high yield and purity of the isolated islets [62]. At this point, the pellet containing acinar and endocrine tissue is collected and assessed for islet count, viability, purity, and endotoxin content, and a sample is sent for Gram stain followed by microbiology culture. This step is very important, because direct infusion of large amounts of pancreatic tissue into the portal vein could lead to increased portal pressure and bleeding, as well as intrahepatic microembolization and inflammation [52,63]. Total pancreatectomy is performed in numerous surgical centers, but there are not many isolation laboratories. In 1997, Rabkin et al. [18] reported a case in which the removed pancreas was sent to a laboratory half a continent distant from the surgical center, and islets were prepared and returned to the original hospital for autotransplantation 16 h after resection. Ten months following transplantation, normal glycemia and insulin independence were noted in this patient. This report showed that the isolation laboratory can be located in the other hospital, distally to the surgical center, which increases accessibility of TPAIT [18].

In a remote islet transplantation, a time of cold ischemia is very important. In 2017, Kesseli et al. [64] published results of their multicenter retrospective study on the impact of cold ischemia on pancreatic islet cell yield. This study included patients undergoing complete or partial pancreatectomy for RAP or CP unresponsive to conservative treatment. Twenty-five patients were analyzed. This study showed that cold ischemia time did not influence islet yield in patients undergoing pancreatectomy with remote isolation [64]. In another report in 2017, Kesseli et al. [65], in their retrospective three-centers cohort study, compared remote and local islet isolation in patients undergoing TPAIT. Two centers performed remote isolation and one performed local isolation. The results of this analysis are very interesting. There was a similar long-term (following 1 year after TPAIT) insulin independence in remote and local groups, but metabolic control was better in patients following the procedure with local islet isolation [65]. In 2012, Khan et al. [25] described a case report of an emergency AIT following a traumatic Whipple pancreaticoduodenectomy and subsequent total pancreatectomy performed on a 21-year-old patient who had been wounded with multiple abdominal gunshot wounds. In the described case, despite long cold ischemia (the pancreas was preserved in UW solution for 9.25 h prior to islet isolation), insulin independence was noted at one and two months following surgery [25].

A microbial status of transplanted islets is a controversial topic. According to most authors, positive cultures do not influence the outcome in patients following TPAIT (including postoperative infectious complications and islet cell viability). In cases of positive cultures, prophylactic antibiotic therapy is generally recommended [66,67,68,69,70]. Colling et al. [68] analyzed retrospectively the sterility cultures from both the pancreas preservation solution used to transport the pancreas and the final islet preparation for intraportal infusion of patients undergoing TPAIT between April 2006 and November 2012. In this study, 251 patients were included. Among them, in 151 (61%) patients, one or more positive bacterial cultures from the pancreas preservation solution or final islet product were reported. In 73 (29%) patients, infectious complications were noted, but only in 7 (4.7%) patients with positive cultures, an infectious complication was caused by the same organism as that isolated from their pancreas or islet cell preparation [68]. Therefore, the authors concluded that there were frequent positives cultures in isolation solutions, but it was associated not frequently with a clinical infectious complication [68]. Gołębiewska et al. [69], in their study including 28 patients, noted bacterial contamination in over 30% of islet preparations. Infectious complications were observed in 50% of patients. Infectious complications were more frequent in patients with positive cultures compared to patients with negative cultures (57% vs. 21%). This finding was significantly associated with the duration of CP (*p* = 0.04). Similarly to Colling’s report [68], these authors did not note an association between pathogens isolated from the pancreas and those identified during the infection [69]. Moreover, Trisler et al. [70] did not find an association between bacterial islet contamination and postoperative outcome in patients undergoing TPAIT. The authors retrospectively analyzed 61 patients. Among them, positive islet cultures were noted in 29 (47.5%) patients. In 23 (79.3%) of these patients, antibiotic prophylaxis was administered. The incidence of postoperative infectious complications was comparable in positive and negative groups (41% vs. 34%, *p* = 0.57). There were no infections in the six islet-culture-positive patients not receiving antibiotics [70].

In contrast to the above-mentioned studies, Trinh et al. [71] noted that infected islets were associated with worse results. In this study, patients with islets suspected to be positive for cultures (*p* = 0.01), and with positive gram stains (*p* = 0.02), were associated with worse procedure results [71]. Jolissaint et al. [72] reported worse results among patients with bacterial contaminants in the final islet preparations. In these patients, a significantly lower islet yield and C-peptide level and no insulin independence following surgery were noted. The authors concluded that although autotransplantation of culture-positive islets was generally safe, it was associated with higher rates of graft failure and poor islet yield [72].

Prepared pancreatic islets are infused into the portal vein via a stump of the splenic vein, or direct puncture of the portal vein, or by cannulation of the umbilical vein. Rafael et al. [73] described intramuscular (to the brachiradialis muscle of the right forearm) islet infusion in a 7-year-old girl with hereditary pancreatitis. The procedure was successful and achieved insulin independence. A control of portal pressure is very important during intraportal islet infusion. Significant tissue volume infused into the portal vein and increased portal pressure can cause a reduction in blood flow leading to portal vein thrombosis [52]. Wilhelm et al. [74] analyzed a large cohort of 233 patients undergoing TPAIT. Based on this analysis, the authors proposed thresholds for pancreatic tissue-volume (TV) for safe intraportal islet autotransplantation following a total pancreatectomy. They recommended TV <0.25 cm^3^ per kg during islet preparation and not to exceed 25 cm H_2_O of portal pressure. In a case of portal pressure higher than this value, a temporary stopping of intraportal infusion is recommended [74].

## 6. Postoperative Monitoring and Management

It should be noted that the clinical effect of islet transplantation is not immediate, because a time for recovery and the engraftment of islets is needed. Therefore, a strict glycemic control is necessary and immediate insulin therapy following surgery. At the beginning, intravenous continuous insulin infusion is recommended followed by subcutaneous administration. Insulin therapy is needed for minimally three months after surgery, and gradual insulin weaning according to glycemia and hemoglobin A1C is recommended [52].

Various markers are measured in patients following TPAIT in order to control the beta-cell function. Gołębiewska et al. [75] analyzed the utility of several markers in postoperative monitoring. The analyzed parameters were as follows: a 90 min serum glucose level, a weighted mean C-peptide in mixed meal tolerance test (MMTT), the Secretory Unit of Islet Transplant Objects (SUITO), the transplant estimated function (TEF), a homeostasis model assessment (HOMA-2B%), a C-peptide/glucose ratio (CP/G), a C-peptide/glucose creatinine ratio (CP/GCr), and a BETA-2 score. A thorough discussion of all these parameters is beyond the scope of this review. This comparative analysis showed that the BETA-2 score was the most useful indicator of beta-cells’ function. The beta score was calculated from the patient’s daily insulin requirements, HbA1C, the fasting plasma glucose level, and stimulated/fasting C-peptide levels [75]. The various other parameters for beta-cell-function monitoring are postulated in the recent literature. Bellin et al. [76] described the novel beta-cell death marker unmethylated insulin (INS) DNA. In the authors’ opinion, persistent posttransplant increased INS DNA correlates with higher hyperglycemia at 90 days [76]. Additionally, various new drugs are investigated in patients following TPAIT. According to some authors, a postoperative inflammatory reaction can lead to a deterioration of beta-cell function. According to this hypothesis, anti-inflammatory drugs should decrease a risk of this complication. Nazirudin et al. [77] reported the use of a combination of etanercept and anakinra (ANA + ETA) to block inflammatory islet injury in 100 patients undergoing TPAIT. The patients were divided into three groups: no treatment (control (CTL)), etanercept alone (ETA), or a combination of etanercept and anakinra (ANA + ETA). Graft function was evaluated by fasting blood glucose, basal C-peptide, the SUITO index, and hemoglobin A1C. The authors noted decreased levels of inflammatory parameters (interleukin-6, interleukin-8, and monocyte chemoattractant protein 1) in patients receiving ANA + ETA compared with the CTL patients. Moreover, an improved beta-cell function confirmed by basal C-peptide, glucose, hemoglobin A1C, and the SUITO index was reported. Insulin requirements were comparable in all groups. This study showed anti-inflammatory and beta-cell function improving of ANA + ETA [77]. In the other study, McEachron et al. [78], in their randomized controlled trial including 83 patients undergoing TPAIT, investigated the protective role of sitagliptin (DPP-4 inhibitor therapy). It is an inhibitor of the enzyme dipeptidyl peptidase-4 (DPP-4) which degrades the gut-derived hormones glucagon-like peptide-1 (GLP-1) and glucose-dependent insulinotropic peptide (GIP), having a protective effect on beta cells. According to the authors’ hypothesis, blocking of DPP-4 should improve the postoperative outcome via increased levels of gut hormones having a protective effect on beta cells. In this study, two groups (patients receiving sitagliptin vs. placebo) were compared. This study showed that although sitagliptin increased GLP-1, it did not improve diabetes outcomes in patients following TPAIT [78]. These studies show that further investigations are needed to find drugs that may improve the postoperative outcome in patients following TPAIT.

## 7. Short-Term Results and Complications Following TPAIT

Various complications during and after TPAIT have been reported in the literature. Portal vein thrombosis (PVT) and bleeding are the most important intraoperative complications. As it was mentioned above, a risk of these complications is associated with portal pressure that is associated with islet volume and purity [52,74,79]. The incidence of PVT has decreased after improving islet preparation and purification. The PVT incidence is about 5% [80]. Robbins et al. [81], in their study including 183 patients, retrospectively assessed the methods of PVT prophylaxis in patients treated at a surgical center of the University of Minnesota. In this experienced surgical center, since 2011, screening abdominal duplex ultrasounds to assess portal venous flow and the presence of PVT after islet autotransplant is used. It is performed routinely on postoperative day 5, or days 1 and 7, depending on the primary surgeon. The prophylaxis schemes were as follows: enoxaparin 40 mg subcutaneously twice daily for 7 days, or unfractionated heparin as a continuous infusion (~5 units/kg/h) until postoperative days 3–5 followed by subcutaneous enoxaparin through day 7. PVT was noted in 12 patients. Anticoagulation therapy (intravenous or subcutaneous) did not influence on the PVT risk. The incidence of PVT was comparable in patients receiving heparin (5%) and enoxaparin (8%) (*p* = 0.5). There was a statistical difference (0% vs. 6%) (*p* = 0.02) regarding other secondary thrombotic complications (not associated with TPAIT) [81].

Postoperative reactive thrombocytosis (RT) (platelets ≥500 K/μL), partially related to splenectomy, has been reported in patients following TPAIT. It also can lead to PVT in the postoperative outcome. In Boucher et al.’s study [82], RT was noted in 93.8% of 42 patients undergoing TPAIT. In order to decrease a risk of thrombotic complications, heparin infusion is used. It can lead to bleeding complications and heparin-induced thrombocytopenia [83].

In a large cohort study by Sutherland et al. [84], including 409 patients, surgical complications requiring relaparotomy were noted in 15.9%. Among them, bleeding (9.5%) was the most common complication. Analyzing the last 230 cases, the frequency of reoperation for bleeding was 7.4%, but was 15% in patients with a postinfusion portal pressure of >25 cm H_2_O, and it was 2 1/2 times higher than the 6% with a lower portal pressure. The other major complications were as follows: anastomotic biliary leak (1.4%), enteric leak (2.8%), intraabdominal infection requiring reoperation (1.9%), wound infections requiring surgical debridement (2.2%), and others such as bowel obstruction, omental infarction, bowel ischemia, and delayed reconstruction because of bowel edema, gastrointestinal tract perforation that required reoperation in 4.7% of patients. In 2 (<1%) patients, reoperation for splenectomy due to ischemic or bleeding preserved spleen was performed. In-hospital mortality was 1.2% (5 patients): due to peritonitis secondary to a perforated colon (1), a pulmonary embolus (1), peritonitis secondary to intestinal perforation caused by a feeding tube (1 child), sepsis (1), and multiple organ failure (1) [84].

Bhayani et al. [85] compared morbidity in patients undergoing TPAIT and total pancreatectomy (TP) alone. There were 126 (40%) patients undergoing TP and 191 (60%) patients undergoing TP + AIT. CP was the most common indication for surgery in both groups. Benign neoplasms were present in 46 (14%) patients. Patients undergoing TPAIT were younger with a lower number of comorbidities compared to the TP group. Despite this, a higher major morbidity rate was noted after TPAIT (79 (41%)) compared to TP (36 (29%)) (*p* = 0.02). The minor morbidity rate (19%) was similar (*p* = 0.965). Moreover, the transfusion rate was higher in patients following TPAIT (39 (20%)) compared to TP (9 (7%)) (*p* = 0.001). A significantly longer duration of operation (530 (370–1007) minutes vs. 310 (75–1140) minutes) (*p* < 0.0001) was reported in the TPAIT group. Moreover, a significantly longer duration of hospitalization was noted in patients undergoing TPAIT (13 days vs. 9 days) (*p* < 0.0001), but the mortality rate was comparable in both groups (1 (1%) vs. 3 (2%) (*p* = 0.147). The frequency of detailed complications (surgical site infection, venous thromboembolism, pneumonia, sepsis, shock) was comparable in both groups. This comparative, multicenter study showed higher major morbidity and transfusion rates, and a longer duration of hospitalization, but comparable mortality in patients following TPAIT compared to TP alone [85].

Shahbazov et al. [86] analyzed risk factors for early readmission of patients following TPAIT. The study involved 83 patients undergoing TPAIT between 2006 and 2014. Twenty-one (25.3%) patients were readmitted within 30 days. In this study, gastrointestinal problems (52.4%) and surgical site infection (42.8%) were the most frequent reasons for readmission. The primary duration of hospitalization and reoperation were risk factors for early readmission. Delayed gastric emptying (DGE) was also a risk factor of readmission. In multivariate analysis, patients undergoing pylorus preserving pancreatectomy were nine times more likely to be readmitted than the antrectomy group (*p* = 0.044). Preoperative parameters (age, gender, previous surgery, previous pancreatic or biliary stenting, alcohol use, smoking, comorbidities, obesity) were similar in readmitted and non-readmitted patients [86].

McSwain et al. [87] analyzed factors influencing postoperative outcomes and the complications rate in patients undergoing TPAIT. The analysis included 161 patients. The morbidity rate was 46.6%. It was associated with hemoglobin levels on postoperative day 1, intraoperative goal-directed treatment, increased intraoperative blood loss, and total amount of intraoperative insulin administration. Duration of hospitalization was significantly associated with the number of complications, use of goal-directed therapy, duration, and postoperative day 1 hemoglobin levels. In patients with a higher number of complications, a significantly longer duration of hospitalization was noted (*p* < 0.001). Additionally, duration of hospitalization in intensive care unit (ICU) was negatively correlated with postoperative hemoglobin levels [87].

Matsumoto et al. [88] analyzed 27 patients undergoing TPAIT and compared islet characteristics and clinical outcomes between patients with complications (complication group) and without (non-complication group), as well as cases with purification (purification group) and without (non-purification group). Tissue volume significantly correlated with maximum (R = 0.61), final (R = 0.53), and delta (i.e., difference between base and maximum; R = 0.71) portal pressure. In patients with complications, a significantly higher body mass index, TV, islet yield, and portal pressure were reported. This observation confirmed the association of complications rate with high TV and high portal pressure. Thus, high tissue volume causes high portal pressure, which results in complications. The complications rate influenced long-term insulin independence. Insulin independence was noted only in 25% of patients with complications who became insulin-free, whereas in 49% of patients without complications. Moreover, a higher islet yield and insulin independence rate was noted in the purification group. The final tissue volume, portal pressure, and complication rates were similar in both groups. In conclusion, high tissue volume was associated with high portal pressure and complications in autologous islet transplantation. Islet purification effectively reduced tissue volume and had no negative impact on islet characteristics. Therefore, in the authors’ opinion, islet purification can reduce the risk of complications and may improve clinical outcomes for autologous islet transplantation when tissue volume is large [88].

Regarding other infrequent complications, in 2015, Bellin et al. [89] described a case of autoimmune-mediated beta-cell failure following TPAIT in a 43-year-old female. In this patient, TPAIT with a high mass islet graft of 6031 islet equivalents (IEQ)/kg was performed. There was no presurgical beta-cell autoimmunity. The type 1 diabetes within the first year after TPAIT led to complete loss of beta-cell function [89].

## 8. Long-Term Results and Quality of Life in Patients Following TPAIT

The goal of TPAIT in patients with RAP and CP is to remove the damaged pancreas, relieve the pain, prevent type 3 diabetes, and improve QoL. Significant pain reduction, narcotics weaning, and insulin independence significantly improve QoL. In a large cohort study by Sutherland et al. [84], 2/3 of patients were alive at 15 years. The survival rate was similar in adult and pediatric patients. In this study, actuarial overall survival (OS) for the entire series of 409 patients was as follows: 97% (1-year survival), 90% (5-years survival), 81% (10-years survival), 66% (15-years survival), and 62% (20-years survival). Regarding insulin dependence, three patient groups were distinguished in this study: insulin independent (II), partial function (PF), insulin dependent (ID). Six months following surgery, there were 61 (23%) II, 155 (58%) PF, and 52 ID (19%). Three years following surgery, there were 51 (30%) II, 55 (33%) PF, and 63 ID (37%). The narcotic use was 97% before operation, 91%—3 months, 61%—6 months, 54%—12 months, and 51%—24 months following surgery. Pancreatitis pain decreased from 97% before operation to 23%—24 months following TPAIT. Thus, a significant decrease in the narcotic use and pain level was observed (*p* < 0.001). Moreover, Health-Related Quality of Life Outcomes (HRQOL) assessed using the SF-36 Survey improved significantly in terms of physical and mental components. Among 53 pediatric patients, the 1-year, 5-years, and 10-years patient survival rates were 98%, 98%, and 79%, respectively. Regarding islet function in this group, during 3 years, there were 55% II, 25% PF, and 20% ID patients [84].

Solomina et al. [90] analyzed long-term results in 20 patients who underwent TPAIT at the University of Chicago with a median follow-up of 28 months (2–38). All 20 patients completed a day 75 follow-up visit; 15 patients completed a 1-year follow-up; and 9 of them completed a 2-year follow-up. This study showed a decrease in the number of patients requiring opioids (from 16 (80%) prior to surgery to 2 (13%) 1 year after surgery). The persistent phantom pancreatic pain was noted in only 1 (6.5%) patient. Moreover, a decrease in drug requirements (from a median 56.3 (0–240) morphine equivalent dose to 5 (0–130) on day 75 and to 0 (0–30) at the 1-year follow-up) was observed. In 5 patients (25%), complete stopping of insulin therapy was possible, maintaining a hemoglobin A1C of 5.9% (5–6.3). In 8 (53%) patients, insulin independence was achieved one year following surgery with a hemoglobin A1C of 6% (5.5–6.8). Both the Physical Component Score (PCS) and Mental Component Score (MCS) improved regardless of changes in insulin requirements [90]. The fact that there is no association between insulin requirements and QoL was also observed in another study by Dorlon et al. [91]. The authors analyzed 74 patients undergoing TPAIT. The authors noted an increase in insulin requirements after surgery (from 5 units/day to 19 units/day) at 6 months, with later increases to 21 units/day at 12 months, and 26 units/day at 2 years. A mean preoperative QoL was 26 for the physical component and 36 for the mental health component, and the postoperative PCS was 33 at 6 months (*p* < 0.001), 36 at 12 months, and 36 at 2 years; the MCS was 42 at 6 months (*p* = 0.007), 41 at 12 months, and 41 at 2 years. Thus, this study revealed no correlation between PCS and MCS QoL and daily insulin requirements (r = −0.016 and r = 0.039, respectively) [91]. The improved QoL was also shown in some other studies [92,93,94,95,96,97,98,99,100,101]. Chinnakotla et al. observed an improved QoL in 80 patients undergoing TPAIT for HGP [31]. Wilson et al. [92] presented a decreased cost and increased QoL in patients with minimally changed chronic pancreatitis (MCCP) following TPAIT compared to patients treated conservatively (USD 153,575/14.9 QoL and USD 196,042/11.5 QoL, respectively). A study by Morgan et al**.** [94] including 127 patients undergoing TPAIT showed an improved QoL in selected CP patients. According to authors, PCS QoL improves quickly after surgery, whereas mental QoL improves gradually. Moreover, opioid abuse can predict physical QoL improvement [94]. Improved QoL was also observed in patients undergoing TPAIT in patients after previous other surgical procedures (such as pancreatic resection or decompression). Wilson et al**.** [95] analyzed 64 patients including 32 (50%) after pancreaticoduodenectomy, 11 (17%) after distal pancreatectomy, 8 (13%) after Frey, 8 (13%) after Puestow, and 5 (8%) after Berne procedures. In all patients, an improved QoL was observed [95]. John et al**.** [96] reported a significant negative correlation between dysmotility scores and SF-12 physical scores (r = −0.46, *p* = 0.008, 95% CI −0.70 to −0.13) and a significant negative correlation between self-reported abdominal pain and both physical and mental SF-12 scores (r = −0.67, *p* < 0.001, 95% CI −0.83 to −0.41 and r = −0.39, *p* = 0.03, 95% CI −0.65 to −0.04). The authors did not report a correlation between gastrointestinal dysmotility and self-reported pain [96].

## 9. TPAIT in Pediatric Patients

CP and RAP are also observed in children. In this specific patients’ group, the above-mentioned hereditary/genetic pancreatitis is frequently observed. The long-term results with decreased insulin requirements, decreased pain, and improved QoL are reported also in pediatric patients following TPAIT. There are numerous studies presenting results of TPAIT in children in the worldwide literature [1,56,57,101,102,103,104,105,106,107,108,109,110,111]. One of the largest studies by Sutherland et al. [84] included 409 patients (53 children) and showed a higher rate of postoperative insulin independence in children (55%) compared to adults (25% of adults). In this observation, all children used opioids before surgery, 39% at follow-up; pain improved in 94%; 67% became pain-free following surgery. Chinnakotla et al. [105] retrospectively analyzed 75 pediatric patients undergoing TPAIT for CP after failed medical, endoscopic, or surgical treatment between 1989 and 2012. In 90% of patients, a decrease in pancreatitis pain was noted (*p* < 0.001), and insulin independence was achieved in 31 (41.3%) children. Insulin independence was correlated with a younger age (*p* = 0.032), lack of a prior Puestow procedure (*p* = 0.018), lower body surface area (*p* = 0.048), higher IEQ/kg of body weight (*p* = 0.001), and total IEQ (100,000 units) (*p* = 0.004). There were three independent factors associated with insulin independence in multivariate analysis: male gender, lower body surface area, and higher total IEQ/kg. The strongest single factor associated with insulin independence was total IEQ (100,000 units) (odds ratio = 2.62; *p* < 0.001) [105]. Berger et al. [109] compared outcomes of 57 positive and 29 negative cultures in pediatric patients undergoing TPAIT for CP. The higher rates of previous pancreas surgery (*p* = 0.007) and endoscopic retrograde cholangiopancreatography (*p* < 0.0001) were noted in patients with positive cultures. There was no association between positive cultures and posttransplant infections (*p* = 1.00) or prolonged duration of hospitalization (*p* = 0.29). Increased rates of graft failure occurred at 2 years posttransplant (*p* = 0.041), but not when adjusted for islet mass transplanted (*p* = 0.39) that was reported in patients with positive cultures [109]. A retrospective study by Bellin et al. [110], including 17 children (9 female) age ≤8 years, showed very good results of TPAIT in young pediatric patients (3–8 years old). Pain relief and opioid cessation were achieved in all children. Hospitalization rates decreased from 5.00 hospitalization episodes per person-year before TPAIT, to 0.35 episodes per person-year after TPAIT. The higher rate of insulin independence (14 of 17 patients, 82%) was noted in children aged ≤8 years compared to an independence rate of 41% in 399 patients older than 8 years of age undergoing TPAIT over the same period (*p* = 0.004). The median postoperative HbA1C was 5.9%, and the mean postoperative HbA1C was ≤6.5% (in all apart from two patients) [110]. In a largest single-center study by Chinnakotla et al. [1], including 581 CP patients (490 adults and 91 children) undergoing TPAIT, better postoperative outcomes were noted in children in comparison to adults. The female percentage was significantly higher in adults compared to children (74.9% vs. 54.9%, *p* < 0.001). Since 2006, over 80% of the pediatric transplants were performed in the past 4 years (*p* = 0.028). So, the number of pediatric procedures has increased recently. The primary cause of CP was different in adults and children: idiopathic (48.8%) in the adults vs. hereditary/genetic in most (68.1%) of the children (*p* < 0.001). Islet yields were higher in children compared to adults (*p* = 0.033). The previous procedures, including surgery, and endoscopic retrograde cholangiopancreatography (ERCP), were more frequently reported in adults. In 151 (26%) patients, previous surgical procedures had failed. The duration of narcotic use (*p* = 0.005) and the duration of either acute pancreatitis or CP (*p* < 0.001) were longer in adults compared to children. During the 37-year observation, 48 patients died (44 adults and 4 pediatric patients). The hospital 30-day mortality was reported in four patients. Higher insulin independence (*p* = 0.002), lower islet graft failure at 1 year (*p* = 0.001), and lower cumulative incidence of persistent pancreatic pain at 1 year (2% vs. 7.5%) were noted in children compared to adults. The opioid use was similar in adults and children at 1 year (*p* = 0.608), although a decrease in narcotic use was quicker in children than in adults [1].

Similar to adults, laparoscopic TPAIT is also performed in children. Berger et al. [57] compared outcomes in 21 children receiving laparoscopic-assisted TPAIT from 2013 to 2015 and 21 children receiving an open procedure from 2011 to 2015. The rate of surgical complications (*p* = 0.35) was comparable in both groups. Duration of operation (*p* = 0.18), duration of hospitalization (*p* = 0.66), blood loss (*p* = 0.96), blood transfusions (*p* = 0.34), and IEQ/kg transplanted (*p* = 0.15) were also similar. Insulin use and opioid use were comparable. The postoperative opioid use during the 2-year follow-up was higher in the laparoscopic group (0% vs. 23.53%, *p* = 0.04). Moreover, patient satisfaction of surgical scar was similar between groups (*p* = 0.26). Therefore, laparoscopic and open approaches were generally associated with similar results in children [57]. It is different from observations among adults undergoing laparoscopic TPAIT. Some authors have reported benefits associated with laparoscopic TPAIT in adults. John et al. [55] reported a shorter duration of hospitalization and a lower rate of delayed gastric emptying (DGE) with prokinetic use in patients undergoing laparoscopic operation compared to the open approach. Another study by Fan et al. [56] presented a shorter operative time, shorter duration of hospitalization, and quicker opioid independence in patients undergoing laparoscopic TPAIT.

## 10. Summary and Conclusions

Currently, indications for TPAIT have been extended. Initially, small duct CP in patients with intractable pain not responding to conservative, endoscopic, or surgical treatment was an indication for this procedure. Although small duct painful CP (including HGP) is still the common indication for TPAIT, this procedure is increasingly performed for other reasons such as: intraductal papillary mucinous neoplasms, neuroendocrine neoplasms, increased anastomotic risk associated with partial pancreatectomy (so-called “high-risk pancreatic stump”), postoperative pancreatic fistula (POPF) (grade C) requiring relaparotomy with a completion pancreatectomy also in patients primarily operated on for malignant pancreatic or peripancreatic neoplasms. Moreover, other rare pancreatic diseases (such as pancreatic AVM and cystic fibrosis) are also indications for TPAIT. Moreover, pancreatic and ampullary cancer and renal cell carcinoma (RCC) metastasis to pancreas as indications for TPAIT have been reported in the literature. Despite these reports, the use of TPAIT in patients with malignancy is still controversial, and currently it is not a standard management. The oncological safety of TPAIT in these patients should be assessed in large, prospective, randomized trials in order to use this treatment method as the standard treatment in patients in malignancy. Current standard indications for total pancreatectomy with autologous islet transplantation are presented in Table 1. Regarding results, TPAIT is effective in adults and children. It allows patients to achieve pain relief, insulin independence or decreased insulin requirements, and an improved QoL. A summary of the most important and largest studies on TPIAT is presented in Table 2 [1,26,31,32,34,84,85,99,112,113,114,115].

## Figures and Tables

**Table 1 jcm-10-02723-t001:** Current indications for total pancreatectomy with autologous islet transplantation.

**Current Indications for Total Pancreatectomy with Autologous Islet Transplantation**
Small duct painful chronic pancreatitis/Recurrent acute pancreatitis
Hereditary/genetic pancreatitis (HGP)
Benign/borderline pancreatic tumors requiring extensive pancreatectomy
“High-risk pancreatic stump“
Pancreatic arteriovenous malformation (AVM)
Pancreatic cystic fibrosis
**Controversial indications for total pancreatectomy with autologous islet transplantation (only case reports)**
Pancreatic and peripancreatic malignant neoplasms requiring total pancreatectomy

**Table 2 jcm-10-02723-t002:** Summary of the most important and largest studies on total pancreatectomy with autologous islet transplantation.

Authors	Year of Publication	Number of Patients	Indications	Results
Dong et al. [34]	2011	Meta-analysis (15 studies)	Chronic pancreatitis Benign tumors	CP: II 27% (1 year), 21% (2 years), 30-day mortality rate 5% Benign tumors: II 24.28% (1 year), 30-day mortality rate 0%
Bramis et al. [113]	2012	Meta-analysis (5 studies)	Chronic pancreatitis	II 46% (5-year)–10% (10-year) QoL poorly reported No evidence for optimal timing of TPAIT
Sutherland et al. [84]	2012	409 (53 children)	Chronic pancreatitis	Survival 96% (adults) and 98% (children) Complications 15.9% II (3 years) 30% Adults II (3 years) 25% Children II (3 years) 55% Pain improvement 85% (94% children) Opioid cessation 59% (67% children) Improved QoL
Bhayani et al. [85]	2013	191 TPIAT 126 TP	Chronic pancreatitis Benign neoplasms	Comparable mortality in TPAIT and TP Higher morbidity and transfusion rates and longer hospitalization in TPAIT
Balzano et al. [26]	2013	41 (17 malignant neoplasms)	Chronic pancreatitis Periampullary tumor PDAC PCN PNET	II 44% Disease-free 13 patients
Wilson et al. [112]	2014	166	Chronic pancreatitis	Perioperative mortality 0% 5-year survival 94.6% NI 55% (1-year), 73% (5-year) II 38% (1-year), 27% (5-year) Improved QoL
Chinnakotla et al. [31]	2014	484	Chronic pancreatitis (including 80 HGP)	Pain-free patients 90% Younger HGP patients Longer CP duration, higher fibrosis score, lower islet yield in HGP Improved QoL
Chinnakotla et al. [1]	2015	581 (91 children)	Chronic pancreatitis	Better outcome in children (higher II and pain rate at 1 year: 2% vs. 7.5%) Improved QoL
Wu et al. [114]	2015	Meta-analysis (12 studies) (677 patients)	Chronic pancreatitis	II 3.72 per 100 person-years 30-day mortality rate 2.1% Last follow-up mortality 1.09 per 100 person-years
Balzano et al. [32]	2016	58	Chronic pancreatitis POPF grade C (21) Neoplasms (PCN *, PNET) of the pancreatic neck (19) High risk pancreatic stump in PD (32) Periampullary tumors PDAC	Major complications 31% Post-islet complications ** 27% II 100% Disease-free 13 patients
Bellin et al. [115]	2019	250 (10-year of follow-up) of total 742 (30 children)	Chronic pancreatitis	10-year survival 72% BMI > 30 kg/m^2^ predicted mortality Pain relief 82% (10 years), 90% (15 years) Narcotic use 50% (5 years), 37% (10 years) II 20% (10 years) Islet IEQ > 4000 strongest prognoctic factor for islet graft function Improved QoL Better islet function in children
Kempeneers et al. [99]	2019	Meta-analysis (15 studies) (1255 patients)	Chronic pancreatitis	II 30% (1 year), II lesser in alcoholic CP NI 15% to 63% (1 year) Improved QoL

II, insulin independence; QoL, quality of life; NI, narcotic independence; HGP, hereditary/genetic pancreatitis; TPAIT, total pancreatectomy with autologous islet transplantation; TP, total pancreatectomy; PDAC, pancreatic ductal adenocarcinoma; PCN, pancreatic cystic neoplasm; PNET; pancreatic neuroendocrine tumor; * PCN included MCN (mucinous cystic neoplasm), SCN (serous cystic neoplasm), IPMN (intraductal papillary mucinous neoplasm), SPN (solid pseudopapillary neoplasm); ** Post-islet complications included: portal vein thrombosis, liver bleeding, perihepatic hematoma, sepsis/bacteremia, gastrointestinal bleeding; BMI, body mass index; IEQ, islet equivalents.

## Data Availability

Not applicable.

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
