# Peer review of "Total Pancreatectomy with Autologous Islet Cell Transplantation—The Current Indications"

_jcm, 2021, doi:10.3390/jcm10122723_

Round 1
Reviewer 1 Report
In this article, the authors reviewed historical aspects, technical aspects, indications as well as outcomes of TPAIT. Their aim is ambitious, since they tried to address several different aspects of this technique. Here reported my main comments.
Major comments
-
In general, this review results more in a description of several different papers than in an organic overview of the history and of the state of the art of this technique. This leads to a very long paper, hard to read and dispersive. I would try to summarize the evidences described and to re-elaborate them in a shorter and more fluent way.
-
Methods section: it is not clear weather authors want to systematically describe the selection process of the articles included in their review or weather they aimed to describe the development of the scientific literature concerning this topic over the years, or both. In the first case, a more detailed description is needed (e.g exact number of articles finally selected from the 409 initially found publications, exact number of articles per 5-yrs period or decade, exact number of articles by type…). In the second case, a narrative description is enough and some details are unnecessary (e.g. last sentence).
-
History of total pancreatectomy
-
in this section results of the first case series of TPAIT on human is described first, than technique in canine models is reviewed. Furthermore, no information about further developments after the 80’s is provided. I would have rather given an overview of the main progresses in a chronological order and covering a wider time period (until nowadays), since going to the next section we find a gap of nearly 30 years.
-
two statements are repeated (line 63-64 and line 77-78) with different references.
-
-
Indications and contraindications for TPAIT:
-
TPAIT is not the only and main treatment of chronic pancreatitis, since other medical (endoscopic) and surgical treatments should be taken in consideration (longitudinal pancreaticojejunostomy, Frey’s procedure, Berger’s procedure, pancreaticoduodenectomy). In general, TPAIT should be considered in patients who have failed other operations or in patients with small duct or minimal change disease. Authors should outline this point.
-
There is increasing evidence that TPAIT can be performed upfront in hereditary and genetic pancreatitis, however stronger evidences are needed to include them in TPAIT indications. “Hereditary / genetic pancreatitis(HGP) is a specific CP kind that is indication for TPAIT.” Authors should give more and more recent references in support of this statement.
-
As described by the authors, TPAIT was recently performed also in patients with malignant diseases and this is considered an extended indication. Besides describing case series, a more detailed overview of the long term results of TPAIT in this subgroup of patients should be given to allow speculations about its safety and appropriateness. Furthermore, I do not think mice models described by Renaud are enough to establish the oncological safety of this technique.
-
-
Pancreatic islet isolation and transplantattion – technique aspects (should be technical aspects): Pancreatectomy can be perfomed en-bloc. I do not understand the following statements: “The pancreatectomy is performed in two steps: Distal pancreactomy (with or without splenectomy) is performed as the first step and pancreaticoduodenectomy is performed as the second one.”. Furthermore partial pancreatectomy can also be performed.
Minor comments
-
The poor-quality of the language significantly compromises the reader comprehension of the text and should therefore be improved. Here some (few) examples:
-
Line 18 and 34: instead of “It leads to the long-life pancreatic exocrine and endocrine insufficiency” I would rather say “It leads to lifelong pancreatic exocrine and endocrine insufficiency”
-
Line 20: “is performed, in order to”, the comma is not needed
-
Line 47: “studie” should be “studies”
-
Singular is and adjective used improperly
-
Line 72: “The most studies were performed on dogs” should be “ most of the studies were performed on dogs”
-
Numerous others
-
Author Response
Dear Reviewer,
Thank you for peer reviewing of our manuscript jcm-1225497, entitled " Total Pancreatectomy with Autologous Islet Cell Transplantation - the Current Indications".
Thank you for your questions and comments. We have fully addressed all the comments and my responses appear below. Our revised work includes corrections according to reviewers’ comments in the text. The changes, made according to reviewers’ comments, are highlighted in red print in the text.
We take this opportunity to express our gratitude to the reviewers for their constructive and useful remarks. Their comments allowed us to identify areas in my manuscript that needed modification.
We also thank you for allowing us to resubmit a revised copy of the manuscript.
We hope that the revised manuscript is now acceptable for publication in Journal of Clinical Medicine.
Yours sincerely,
Beata Jabłońska.
Reviewer 1
In this article, the authors reviewed historical aspects, technical aspects, indications as well as outcomes of TPAIT. Their aim is ambitious, since they tried to address several different aspects of this technique. Here reported my main comments.
Major comments
- In general, this review results more in a description of several different papers than in an organic overview of the history and of the state of the art of this technique. This leads to a very long paper, hard to read and dispersive. I would try to summarize the evidences described and to re-elaborate them in a shorter and more fluent way.
Answer: The aim of our paper was to review and describe the most important articles on TPAIT. The most essential issues have been more exactly described. So, it is a descriptive, not systematic review, divided into several topic sections.
- Methods section: it is not clear weather authors want to systematically describe the selection process of the articles included in their review or weather they aimed to describe the development of the scientific literature concerning this topic over the years, or both. In the first case, a more detailed description is needed (e.g exact number of articles finally selected from the 409 initially found publications, exact number of articles per 5-yrs period or decade, exact number of articles by type…). In the second case, a narrative description is enough and some details are unnecessary (e.g. last sentence).
Answer: It is not systematic review. According to your suggestion, a narrative description has been presented, without other details as follows:
We have reviewed a PubMed database for the following search terms and meSH terms: total pancreatectomy with / and pancreatic islet transplantation, pancreatic islet autotransplantation, chronic pancreatitis, hereditary genetic pancreatitis, intraductal papillary mucinous neoplasm from 1975 to 2021. We have selected English, full-text publications which exactly corresponded to the subject of our interest. We have noted that most of the studies were performed on dogs. Subsequently, first case reports were described followed by case series, retrospective observational, control, and prospective studies. Initially, singular articles related TPAIT have been published, and the number of publications has increased gradually.
- History of total pancreatectomy
- in this section results of the first case series of TPAIT on human is described first, than technique in canine models is reviewed. Furthermore, no information about further developments after the 80’s is provided. I would have rather given an overview of the main progresses in a chronological order and covering a wider time period (until nowadays), since going to the next section we find a gap of nearly 30 years.
Answer: The aim of this section was to present the first TPAIT in animal and human models. According to your suggestion, the first TPAIT on canine models has been more exactly desribed and an overview of the main progresses in a chronological order until nowadays as follows:
The first successul case of intrasplenic TPAIT performed in canine models was desribed by Mirkowith et al in 1976 [2]. In this experimental study, a dispersed pancreatic tissue was prepared by collagenase digestion without separation of exocrine and endocrine pancreatic components. This unpurified pancreatic tissue was infused into the dogs’ spleen. In 20 out of 25 dogs, a normalized serum glucose was reported [2]. The first report of human TPAIT was published by Najarian, Sutherland and co-authors from the University of Minnesota in 1980 [3]. The authors applied the technique of TPAIT, previously used in dogs, to ten patients with chronic pancreatitis (CP) with small pancreatic ducts and severe, intractable pain. Subtotal pancreatectomy (>95% of the pancreas volume) was performed and the removed pancreas was minced, dispersed by collagenase digestion and infused into the portal vein < 2 1/2 hours after pancreatic resection. The pain relief was achieved in all patients. One patient died of a complication not related to this procedure. In the remaining nine patients, insulin independence or the significant decrease of insulin requirement was noted. The proper function of transplantated pancreatic islets was confirmed by C-peptide investigation [3]. In the first years, most publications on TPAIT regarded studies carried out on animal models (dogs, pigs, rats). The most studies were performed on dogs. The pancreatic islets were autotransplanted either directly into the splenic pulp, or into the splenic artery or the portal vein [4-11]. In 1978, based on studies performed on canine models, Kretschmer et al [5] concluded that a direct implantation of pancreatic islets into the splenic pulp was better than transplantation into the portal vein or splenic artery because the splenic circulation was not disturbed, portal hypertension was avoided, as well as a postoperative glucose control was better. The next reports on human TPAIT were published in the 80s years of the 20th century [12,13]. Initially, the studies involved patients operated on for intractable pain from CP. In 1990, Farney et al [14] described results of the study including 26 patients who underwent autotransplantation of dispersed pancreatic islet tissue combined with total or near-total pancreatectomy for treatment of CP between 1977 and 1991. Pancreatic islets were injected intraportally in 22 patients and into the renal subcapsule in two patients. In the next years, the pioneer surgical center at the Minnesota University has evaluated. In 1995, the athors published results of islet autotransplantation performed in 48 patients following total/near-total (> 95%) pancreatectomy (n=43) and partial pancreatectomy (n=5) between 1977 and 1995. The authors showed that slet autotransplantation could prevent long-term diabetes in more than 33% of patients [15]. Also, in 1995, a case of pregnancy following TPAIT was reported. This case report confirmed a high QoL in patients following TPAIT [16]. In 1996, a case of a 12-year-old boy who underwent successful TPAIT for intractable pain caused by idiopathic CP was reported. Postoperatively, complete relief of abdominal pain was achieved. The patient remained insulin-independent, with normal fasting blood glucose and hemoglobin A1C levels, for 21/2 years [17]. In 1997, a case of transcontinental shipping of pancreatic islets for autotransplantation after total pancreatectomy was reported. Pancreatectomy is performed at most hospitals but islets are prepared at only a few centers. Therefore, the possibility of successful transcontinental shipping of pancreatic islets for autotransplantation after total pancreatectomy is very important in order to spread this procedure [18]. The first islet autotransplant programme in the United Kingdom (the Leicester experience) and the first series in the world to use the spleen as a site for the islet graft was reported in 1999. Over an 11 month period, 7 patients underwent TPAIT for CP. In 6 patients, islets were embolized into the liver via the portal vein (median transplanted volume=8.5 ml). Three patients received islets into the splenic sinusoids via a short gastric vein. One patient received islets into the spleen alone [19]. In 2001, good long-term results of intrahepatic autoislet transplantation in six CP patients with stable beta-cell function and normal levels of blood glucose and HbA1c for up to 13 years [20]. In 2003, the authors from the Leicester general hospital described the results of TPAIT performed in 40 patients with CP. The follow-up times ranged from 6 months to 7 years [21]. In 2008, the long-term results of 46 patients were published [22]. Also, in 2003, the authors from the University of Cincinnati College of Medicine published the results (22 patients) [23]. In 2004, a case report of TPAIT with the use of a temporarily exteriorized omental vein was reported by the authors from the Minnesota University [24]. The indications for TPAIT have been extended. In 2012, an emergency autologous islet transplant after a traumatic Whipple operation and subsequent total pancreatectomy performed for a 21-year-old patient who was wounded with multiple abdominal gunshot wounds was reported [25]. In 2013, Balzano et al described extended indications for islet autotransplantation to grade C pancreatic fistula (treated with completion or left pancreatectomy, as indicated); total pancreatectomy as an alternative to high-risk anastomosis during pancreaticoduodenectomy; and distal pancreatectomy for benign/borderline neoplasm of pancreatic body-neck [26]. Also, the procedure approach has been modified through the years. In 2012, the first robot-assisted pancreatoduodenectomy with preservation of the vascular supply for autologous islet cell isolation and transplantation was reported. So far, both open and robotic-assisted TPAIT are performed [27].
- two statements are repeated (line 63-64 and line 77-78) with different references.
Answer: It has been precised as follows:
The next reports on human TPAIT were published in the 80s years of the 20th century [12,13].
- Indications and contraindications for TPAIT:
- TPAIT is not the only and main treatment of chronic pancreatitis, since other medical (endoscopic) and surgical treatments should be taken in consideration (longitudinal pancreaticojejunostomy, Frey’s procedure, Berger’s procedure, pancreaticoduodenectomy). In general, TPAIT should be considered in patients who have failed other operations or in patients with small duct or minimal change disease. Authors should outline this point.
Answer: It has been written that in the section 4, lines 3-7 in the primary version, and according to your suggestion some additional informations have been outlined as follows:
The authors recommend TPAIT in patients in whom previous conservative, endoscopic, and surgical treatment was not effective. Therefore, patients with small duct CP are main candidates for this treatment method. In these patients, the main goal of TPAIT is to relieve the pain and to improve QoL (by total pancreatectomy) as well as to prevent postoperative type 3 diabetes (by autologous islet transplantation) [28]. In the authors‘ opinion, severity, frequency, and duration of pain, requirement of narcotic drugs, decreased QoL, residual pancreatic islet function, rate of disease progression, and patient’s age are important in the choice of optimal timing of the procedure [28]. It should be pointed that in CP, endoscopic and surgical treatments should be taken in consideration. Endoscopic treatment (ET) is the most useful in patients with pancreatic duct lithiasis, obstruction and dilation. It should be the first-line option because it is less invasive than surgery. Surgery such as drainage operations (Puestow’s, Partington-Rochelle’s, Duval’s procedures), resectional operations (partial and subtotal or total pancreatectomies), resections with extended drainage (Beger’s, Frey’s procedures) should be the first-line option in patients in whom ET has failed or in those with a pancreatic mass with suspicion of malignancy. In general, TPAIT should be considered in patients who have failed other operations or in patients with small duct or minimal change disease [29].
- There is increasing evidence that TPAIT can be performed upfront in hereditary and genetic pancreatitis, however stronger evidences are needed to include them in TPAIT indications. “Hereditary / genetic pancreatitis(HGP) is a specific CP kind that is indication for TPAIT.” Authors should give more and more recent references in support of this statement.
Answer: The paragraph regarding HGP as indication for TPAIT has been extended using more and more recent referenes as follows:
HGP is caused by a mutation in the gene encoding cationic trypsinogen (protease serine 1, or PRSS1), mapped to the 7q35 region of the long arm of human chromosome 7. Pathogenic mutations in the PRSS1 are associated with more than an 80% chance for recurrent acute and/or chronic pancreatitis, as well as exceptionally high lifetime risk (estimated at 40%) for pancreatic ductal adenocarcinoma. Because of a high cancer risk, these patients distinct from the general TPAIT population. TPAIT in these patients is considered typically in young patients at first symptoms (often <10 years) and higher lifetime risk for pancreatic cancer. Patients with PRSS1-mediated CP are increasingly being considered for TPIAT when pain and disability are severe. So, in HGP patients, the goal of TPAIT is not only to relieve a pain and increase QoL, but also to decrease a pancreatic cancer risk. A strong association between HGP duration and the diabetes outcomes has been observed. In comparison of HGP patients with those patients with no known genetic risk factors, a smaller and more fibrotic pancreas, lower islet mass, and lower probability of ever achieving insulin independence was noted in HGP patients compared to non-genetic CP [30]. A study by Chinnakotla et al [31] including 80 HGP patients showed that TPAIT in HGP patients cause provide long-term pain relief (90%) and preservation of β-cell function. HGP patients with a high lifetime risk of pancreatic cancer should be considered earlier for TP-IAT before pancreatic inflammation results in a higher degree of pancreatic fibrosis and islet cell function loss. Therefore, these patients are typically younger compared to the patients undergoing TPAIT for other indications [31]. A study by Bellin et al including 64 HGP patients showed that postoperative outcome was adversely impacted by older age and prolonged disease. In particular, islet mass was lower and risk of diabetes high in older patients with prolonged disease. This should be considered during counseling this subgroup of TPIAT recipients for surgical management [30].
- As described by the authors, TPAIT was recently performed also in patients with malignant diseases and this is considered an extended indication. Besides describing case series, a more detailed overview of the long term results of TPAIT in this subgroup of patients should be given to allow speculations about its safety and appropriateness. Furthermore, I do not think mice models described by Renaud are enough to establish the oncological safety of this technique.
Answer: The paragraph regarding malignant diseases as indications for TPAIT has been modified, and a more detailed overview of the long term results of TPAIT in this subgroup of patients has been presented as follows:
Clinical contraindications for pancreatectomy with autologous islet transplantation, including pancreatic neoplasms, are controversial. There are reports showing the successful use of this procedure in patients with pancreatic neoplasms (including IPMN, insulinoma, pancreatic and ampullary cancer, renal cell carcinoma (RCC) metastasis to pancreas) in the literature [33-52]. Ris et al [34] described 25 patients undergoing extended pancreatectomy and islet autotransplantation for benign pancreatic tumors. There were 14 patients with benign tumors located in the pancreatic isthmus including 10 (4 mucinous and 6 serous) cystadenomas and four neuroendocrine tumors (3 insulinomas and 1 nonsecreting neuroendocrine tumor) in this cohort. Malignancy was ruled out in each tumor, both based on the intraoperative and postoperative histological findings. The actuarial 10-year survival rate was 100% [16]. Oberholzer et al [35] described the first cases of extended left pancreatectomy with autologous islet transplantation. The authors considered that the risk of tumor dissemination by this procedure was low because of the presence of a single tumor, intraoperative frozen sections assuring tumor-free resection margins, as well as no invasion of surrounding tissue, but there was no information regarding long-term results in these patients [35]. Zureikat et al [38] described ten robotic-assisted TPAIT in patients with indications including IPMN (n=6), pancreatic adenocarcinoma (n=1). In this study, only short-term results were reported [38]. Balzano et al [32] analyzed results of TPAIT performed for extended indications than in the Milan Protocol, including treatment for severe pancreatic fistulas (n = 21), extensive distal pancreatectomy for neoplasms of the pancreatic neck (n = 19) or pancreaticoduodenectomy because of the high risk of pancreatic fistula (n = 32). In this study, in 31 of 58 patients malignant pancreatic or periampullary neoplasms were reported, and ex novo liver metastases following surgery were noted in only 3 patients (median follow-up 914 ± 382 days) [32]. Relapse was observed only in 5 (12.9%) of 31 patients who were disease free after surgery. In comparison of overall survival and disease/progression-free survival of patients with ductal carcinoma treated with islet autotransplantation to those of patients with ductal carcinoma who had surgery in the same period of time but did not receive islet autotransplantation, patients undergoing islet autotransplantation had a better survival than did patients without islet autotransplantation at a similar stage of disease. In fact, malignancy has been considered an absolute contraindication for islet autotransplantation because of the risk to disseminate cancer cells through the infusion of islets, which may still contain some exocrine cells even after purification. Therefore, prospective randomized trials are needed to assess the long-term oncological results of TPAIT [39]. There are a few reports on the use of islet autotransplantation in patients with a malignant neoplasm such as ampullary adenocarcinoma (T3N1M0) with postoperative complications (1 year of follow-up with no cancer recurrence) [40], ampullary adenocarcinoma pT1N0MX G1 with postoperative complications (3 months of follow-up with no cancer recurrence) [41], pancreatic cancer with postoperative complications (1 year of follow-up with no cancer recurrence) [42], pancreatic cancer (T3pN1G2) with postoperative complications (died due to cancer recurrence 2.5 years following surgery) [43]. Kocik et al [46] presented five patients undergoing completion pancreatectomy with autologous islet transplantation as treatment of POPF following pancreaticoduodenectomy due to malignant neoplasms including pancreatic adenocarcinoma, cholangiocarcinoma, and papilla of Vater cancer. Two of five patients were still alive 9 and 28 months after surgery. Savari et al [50] presented a case of TPAIT in a 54-year-old male with an invasive 2-cm ampullary adenocarcinoma (T3N0M0) and splenic vein thrombosis, pseudocyst, and abscess in the pancreatic tail. Six months after surgery, disseminated metastatic disease (adenocarcinoma of the skin in the scar at the site of drain placement in the left upper abdomen and pulmonary metastases) was noted. Gala-Lopez et al [51] reported a case of TPAIT in a 70-year-old female with multifocal pancreatic metastases from RCC. In this patient, insulin independence and no tumor recurrence were noted one year following surgery. Renaud et al [47] published results of the experimental study including patients requiring pancreatectomy for pancreatic ductal adenocarcinoma (PDAC). Immunocompromised mice were transplanted with pancreatic cells isolated from the nonmalignant part of the surgical specimen (experimental group). Results were compared with pancreatic tumor implants (control group). Pancreatic grafts were explanted at 6 weeks for histological analyses. This study included 9 patients and 31 mice were transplanted. In the experimental group, explants were microscopically devoid of tumor cell, and no metastasis was observed. In the control group, all explants were composed of tumor. In this study, the authors demonstrated the absence of local and distant spreading of malignant cells after pancreatic islets xenograft isolated from PDAC patients. In authors’ opinion, these results confirmed oncological safety of TPAIT as valuable alternative to partial pancreatectomy for PDAC patients with a high risk of POPF [47]. Despite of these above mentioned findings, the oncological safety of TPAIT for PDAC patients should be assessed by prospective randomized trials in order to recommend this procedure routinely.
Sakata et al [44] presented TPAIT as treatment of pancreatic arteriovenous malformation (AVM) performed in 3 males in age 39, 52, and 59 years, achieving good results in two of them. Wang et al [45] reported a case of successful TPAIT performed in a 16-year-old boy with intractable pain due to CP coexisting with ulcerative colitis and primary sclerosing cholangitis (PSC). Desai et al [48] described the first report of successful TPAIT for pancreatic cystosis in a 21-year-old woman with pancreatic cystic fibrosis, with insulin independence 11 months following surgery. St Onge et al [49] presented TPAIT in a 4-year-old female with pancreatic cystic fibrosis, with insulin independence one year following surgery.
Concluding, small duct CP with an intractable pain is a common indication for TPAIT. Recently, primary indications for TPAIT have been extended and include selected patients with benign pancreatic tumors, and other rare pancreatic diseases (such as pancreatic AVM and cystic fibrosis). Currently, some primary contraindications (such as IPMN, endocrine tumors) became indications for TPAIT. The use of TPAIT in malignant pancreatic and peripancreatic neoplasms is controversial and currently it is not a standard management. The large randomized trials that are needed to assess long-term oncological results (such as disease-free survival).
- Pancreatic islet isolation and transplantattion – technique aspects (should be technical aspects): Pancreatectomy can be perfomed en-bloc. I do not understand the following statements: “The pancreatectomy is performed in two steps: Distal pancreactomy (with or without splenectomy) is performed as the first step and pancreaticoduodenectomy is performed as the second one.”. Furthermore partial pancreatectomy can also be performed.
Answer: It has been corrected as follows:
The standard procedure preceding islet transplantation involves total or partial pancreatectomy (in two steps or en-block, with or without splenectomy).
Minor comments
- The poor-quality of the language significantly compromises the reader comprehension of the text and should therefore be improved. Here some (few) examples:
- Line 18 and 34: instead of “It leads to the long-life pancreatic exocrine and endocrine insufficiency” I would rather say “It leads to lifelong pancreatic exocrine and endocrine insufficiency”
- Line 20: “is performed, in order to”, the comma is not needed
- Line 47: “studie” should be “studies”
- Singular is and adjective used improperly
- Line 72: “The most studies were performed on dogs” should be “ most of the studies were performed on dogs”
- Numerous others
Answer: English has been improved.

Reviewer 2 Report
Journal of Clinical Medicine, June 2021
Total Pancreatectomy with Autologous Islet Cell Transplantation - the Current Indications
Summary:
In the review the authors described and summarised the indications and the results of Total Pancreatectomy with Autologous Islet Transplantation (TPAIT). Therefor they analysed almost 100 papers from 1975 to 2021.
Broad comments:
The whole paper gives the impression that it is not revised very carefully.
English has to be improved, there are several typing errors and unusual phrasings.
Referencing is a little bit of a mess. Some references do not appear in the text, author names and the numbers in some citations within the text do not correlate. Please revise the references.
Specific comments:
page 2, line 46: 70s not 70th
page 3, line 53: what are the exact search and meSH terms.
page 3, line 57: change “and a number of” to ”and the number of”.
page 3, line 63: why do the references start with number 2? The first appearance of reference 1 is only at page 19.
page 4, line 81: “leads to QoL”, low, high?
page 4, line 88 and 91: “ref 12” is probably wrong.
page 5, line 126: what about references 18, 19 and 20?
page 9, line 169: “performe das”
Please correct.
page 10, line 248:”They recommended TV <0.25cc/kg during islet preparation”
What is meant by that, it doesn’t make any sense.
page 11, line 266: “There various other parameters” ?
page 11, line 267: “funtion”
Please correct.
page 11, line 279: “reuirements”
Please correct.
page 12, line 285-: “blocking of DPP 4 should improved postoperative outcome and via increased levels of protective for beta cells gut hormones.”
Phrasing?
page 12, line 287: “Thse”
Please correct.
page 12, line 293: “rosk”
Please correct.
page 13, line 318: “others (ileus, omental infarction, bowel ischemia (4.7%) bowel obstruction, omental infarction, bowel ischemia or edema, gastrointestinal tract perforation).”
Where does the 4.7% belong to?
page 13, line 328 – 335: What is major morbidity rate and minor morbidity rate besides other complications?
page 14, line 358: “The significant positive correlation between TV and maximum (R=0.61), final (R=0.53), and difference (R=0.71) portal pressure.”
Phrasing?
page 15, line 362: “Insulin independence was noted only in 25% of patients with complications became insulin free, whereas in 49% of patients without complications.”
Phrasing?
page 15, line 363-365: “Moreover, a higher islet yield and insulin independence rate was noted in the purification group. The final tissue volume, portal pressure, and complication rates were similar in both groups. This study showed that islet purification could decrease the risk of complications”.
How can it be, that islet purification could decrease the risk of complications if both group show similar complication rates?
page 15, line 378 – 381: “ 97% (1 year) , 90% (5 years), 81% (10 years), 66% (15 years ), and 62% (20 years). Regarding isnulin depenedence, three patients groups were distinguished in this study: insulin independent (II), partial function (PF), insulin dependent (ID). Six months following surgery, there were 61(23%) II, 155 (58%) PF, and 52 (19%). Three years following surgery, there were 51(30%) II, 55 (33%) PF, and 63 (37%).”
How can it be, that the five year survival is 90% but after three years there are only 63% of the patients are analysed (six months: 268 patients (61+155+52); three years: 169 patients (51+55+63) => 63%)? Please correct also “isnulin depenedence”.
page 16. line 391 – 395: “(from 16 (80%) prior to surgery to 2 (13%) 1 year after”
The percentage in parenthesis is sometimes based on 20 patients sometimes on 15 patients, it’s not clear why.
page 16, line 402: ”ans postoperative PSC”
Please correct.
page 16, line 407: ”increased QoL in patients with in patients with minimally changed”
Please correct
page 16, line 408: “OoL“
Please correct.
page 16, line 409: “Morgan et al [74]“
Morgan is ref 75, where should ref 74 be mentioned in the text? Please correct.
page 16, line 410 and 411: “ QOL “
Please correct.
page 17, line 413 - 420: “Wilson et al analyzed .... improved QoL was observed [77]. John et al [78] reported ... and self reported pain [78] “
Insert ref number after Wilson et al, Wilson is not 77 but 76, John not 78 but 77, Please correct.
page 17, line 434 -437: “ total IEQ (100,000) “
What does this mean, only if there are exactly 100,000 IEQ?
page 18, line 437: “Berger et al [89] compared“
Berger is ref 92. Where are refs 89, 90, and 91 in the text?
page 18, line 447: “insulin independence (n=14.82%) was“
n=14.82% ? Please correct.
page 18, line 450: “Chinnakotla et al [1], “
Why ref 1, never mentioned before.
page 18, line 461: “Gigher insulin independence“
Please correct.
page 19, line 468: “hospitalization (p=0.66). blood loss“
Please correct.
page 19, line 474: “Singh et al [33]“
Wrong name.
page 19, line 476: “Fan et al presented“
Please insert ref number as before.
page 20, line 495: “[1,15,17,64,72, 80, 96-100]“
Ref number 65 is missing.
page 21, line 500: “BMI, body mass index; EIQ, equivalents;“
IEQ not EIQ, equivalents of what?
page 27, line 653: “10.1111/ajt.12354. Epub 2013 Jul 16. PMID: 23859047.Rickels MR, Robertson RP. Pancreatic Islet Transplantation in Humans: Recent“
Rickels MR is a new ref, number? Please correct.
page 28, line 683 and 688: Ref 35 and ref 36 are identical.
page 33, line 814: Ref 59 not mentioned in text.
page 38 and 39, line 952 - 973: Refs 83, 84,85 and 86 are not mentioned in the text.
Author Response
Dear Editor,
Dear Reviewer,
Thank you for peer reviewing of our manuscript jcm-1225497, entitled " Total Pancreatectomy with Autologous Islet Cell Transplantation - the Current Indications".
Thank you for your questions and comments. We have fully addressed all the comments and my responses appear below. Our revised work includes corrections according to reviewers’ comments in the text. The changes, made according to reviewers’ comments, are highlighted in red print in the text.
We take this opportunity to express our gratitude to the reviewers for their constructive and useful remarks. Their comments allowed us to identify areas in my manuscript that needed modification.
We also thank you for allowing us to resubmit a revised copy of the manuscript.
We hope that the revised manuscript is now acceptable for publication in Journal of Clinical Medicine.
Yours sincerely,
Beata Jabłońska.
Reviewer 2
Total Pancreatectomy with Autologous Islet Cell Transplantation - the Current Indications
Summary:
In the review the authors described and summarised the indications and the results of Total Pancreatectomy with Autologous Islet Transplantation (TPAIT). Therefor they analysed almost 100 papers from 1975 to 2021.
Broad comments:
The whole paper gives the impression that it is not revised very carefully.
English has to be improved, there are several typing errors and unusual phrasings.
Referencing is a little bit of a mess. Some references do not appear in the text, author names and the numbers in some citations within the text do not correlate. Please revise the references.
Answer: English has been improved. References have been carefully revised. The references numbers have been changed due to insertion of additional references according to the second Reviewer’s suggestions.
Specific comments:
page 2, line 46: 70s not 70th
Answer: It has been corrected as 70s.
page 3, line 53: what are the exact search and meSH terms.
Answer: The additional search and meSH terms has been added as follows: pancreatic islet autotransplantation; chronic pancreatitis, hereditary genetic pancreatitis, intraductal papillary mucinous neoplasm
page 3, line 57: change “and a number of” to ”and the number of”.
Answer: It has been changed to to ”and the number of”.
page 3, line 63: why do the references start with number 2? The first appearance of reference 1 is only at page 19.
Answer: The number [1] has been added in the introduction.
page 4, line 81: “leads to QoL”, low, high?
Answer: It has been been added as „low”.
page 4, line 88 and 91: “ref 12” is probably wrong.
Answer: It has been corrected as [28].
page 5, line 126: what about references 18, 19 and 20?
Answer: The references have been revised and changed. They regard the use of TPAIT in benign and malignant neoplasms and they are cited (currently as references 35,36,37) in the sentence:
There are reports showing the successful use of this procedure in patients with pancreatic neoplasms (including IPMN, insulinoma, pancreatic and ampullary cancer, renal cell carcinoma (RCC) metastasis to pancreas) in the literature [33-52].
page 9, line 169: “performe das”
Please correct.
Answer: This paragraph has been modified and it has been deleted according to the second Reviewer’s suggestion.
page 10, line 248:”They recommended TV <0.25cc/kg during islet preparation”
What is meant by that, it doesn’t make any sense.
Answer: It means a tissue volume <0.25 cm3 (cubic centimeter) per kg. Abbreviation „cc” has been used in the cited article. Now, it has been changed to <0.25 cm 3 per kg for better understanding.
page 11, line 266: “There various other parameters” ?
Answer: It has been changed as „the various other parameters”
page 11, line 267: “funtion”
Please correct.
Answer: It has been corrected.
page 11, line 279: “reuirements”
Please correct.
Answer: It has been corrected.
page 12, line 285-: “blocking of DPP 4 should improved postoperative outcome and via increased levels of protective for beta cells gut hormones.”
Phrasing?
Answer: This sentence has been rephrased as follows:
According to authors‘ hypothesis, blocking of DPP-4 should improve postoperative outcome via increased levels of gut hormones having a protective effect on beta cells.
page 12, line 287: “Thse”
Please correct.
Answer: It has been corrected.
page 12, line 293: “rosk”
Please correct.
Answer: It has been corrected.
page 13, line 318: “others (ileus, omental infarction, bowel ischemia (4.7%) bowel obstruction, omental infarction, bowel ischemia or edema, gastrointestinal tract perforation).”
Where does the 4.7% belong to?
Answer: It has been precised as follows:
such as bowel obstruction, omental infarction, bowel ischemia, delayed reconstruction because of bowel edema, gastrointestinal tract perforation that required reoperation in 4.7% of patients
Answer: It has been corrected.
page 13, line 328 – 335: What is major morbidity rate and minor morbidity rate besides other complications?
Answer: The authors of the cited article presented the results regarding major morbidity and mortality rate (the types of minor and major complications were not mentioned). The complications such as surgical site infection, venous thromboembolism, pneumonia, sepsis, shock were separately mentioned and described. The major/minor morbidity include mentioned above complications. To show it, the word „other” was changed to ‘detailed’ as follows:
The frequency of detailed complications (surgical site infection, venous thromboembolism, pneumonia, sepsis, shock) was comparable in both groups.
page 14, line 358: “The significant positive correlation between TV and maximum (R=0.61), final (R=0.53), and difference (R=0.71) portal pressure.”
Phrasing?
Answer: This sentence has been rephrased as follows:
Tissue volume significantly correlated with maximum (R = 0.61), final (R = 0.53), and delta (i.e., difference between base and maximum; R = 0.71) portal pressure.
page 15, line 362: “Insulin independence was noted only in 25% of patients with complications became insulin free, whereas in 49% of patients without complications.”
Phrasing?
Answer: This sentence has been rephrased as follows:
Only 1 of 4 patients (25%) in the complication group became insulin free, whereas 11 of 23 patients (49%) in the non-complication group became insulin free.
page 15, line 363-365: “Moreover, a higher islet yield and insulin independence rate was noted in the purification group. The final tissue volume, portal pressure, and complication rates were similar in both groups. This study showed that islet purification could decrease the risk of complications”.
How can it be, that islet purification could decrease the risk of complications if both group show similar complication rates?
Answer: This sentence regarded the association between large tissue volume and high portal pressure which results in complications. In the cited study, authors concluded that high tissue volume was associated with high portal pressure and complications in autologous islet transplantation. Islet purification effectively reduced tissue volume and had no negative impact on islet characteristics. Therefore, in authors’ opinion, islet purification can reduce the risk of complications and may improve clinical outcome for autologous islet transplantation when tissue volume is large. This explanation had been added in the manuscript as follows:
In patients with complications, a significantly higher body mass index, TV, islet yield, and portal pressure were reported. This observation confirmed association of complications rate with high TV and high portal pressure. Thus, high tissue volume causes high portal pressure, which results in complications. The complications rate influenced on long-term insulin independence. Insulin independence was noted only in 25% of patients with complications became insulin free, whereas in 49% of patients without complications. Moreover, a higher islet yield and insulin independence rate was noted in the purification group. The final tissue volume, portal pressure, and complication rates were similar in both groups. In conclusion, high tissue volume was associated with high portal pressure and complications in autologous islet transplantation. Islet purification effectively reduced tissue volume and had no negative impact on islet characteristics. Therefore, in authors’ opinion, islet purification can reduce the risk of complications and may improve clinical outcome for autologous islet transplantation when tissue volume is large
page 15, line 378 – 381: “ 97% (1 year) , 90% (5 years), 81% (10 years), 66% (15 years ), and 62% (20 years). Regarding isnulin depenedence, three patients groups were distinguished in this study: insulin independent (II), partial function (PF), insulin dependent (ID). Six months following surgery, there were 61(23%) II, 155 (58%) PF, and 52 (19%). Three years following surgery, there were 51(30%) II, 55 (33%) PF, and 63 (37%).”
How can it be, that the five year survival is 90% but after three years there are only 63% of the patients are analysed (six months: 268 patients (61+155+52); three years: 169 patients (51+55+63) => 63%)? Please correct also “isnulin depenedence”.
Answer: These results are reported in the cited study. We have checked it and the authors presented such actuarial overall survival (OS) for the entire series of 409 patients (the numbers are correct) and they distinguished three groups according to insulin dependence counted as cited. It could be associated with available information regerding OS from all 409 patients, but insulin requirement assessment was not possible to be performed in all patients. Therefore, it has been corrected as follows:
In this study actuarial overall survival (OS) for the entire series of 409 patients was as follows: 97% (1 year survival), 90% (5 years survival), 81% (10 years survival), 66% (15 years survival), and 62% (20 years).
“isnulin depenedence” has been corrected as „insulin dependence”.
page 16. line 391 – 395: “(from 16 (80%) prior to surgery to 2 (13%) 1 year after”
The percentage in parenthesis is sometimes based on 20 patients sometimes on 15 patients, it’s not clear why.
Answer: It is associated with a decreased number of patients during follow-up. All 20 patients completed day 75 follow-up visit, 15 patients completed 1-year follow-up, and 9 of them completed 2-year follow-up. It has been added in the manuscript in order to explain it as follows:
Solomina et al [90] analyzed long-term results in 20 patients who underwent TPAIT at the University of Chicago during median follow-up was 28 months (2-38). All 20 patients completed day 75 follow-up visit, 15 patients completed 1-year follow-up, and 9 of them completed 2-year follow-up.
page 16, line 402: ”ans postoperative PSC”
Please correct.
Answer: It has been corrected as „PCS” (Physical Component Score).
page 16, line 407: ”increased QoL in patients with in patients with minimally changed”
Please correct
Answer: It has been corrected.
page 16, line 408: “OoL“
Please correct.
Answer: It has been corrected.
page 16, line 409: “Morgan et al [74]“
Morgan is ref 75, where should ref 74 be mentioned in the text? Please correct.
Answer: It has been corrected as [94].
page 16, line 410 and 411: “ QOL “
Please correct.
Answer: It has been corrected.
page 17, line 413 - 420: “Wilson et al analyzed .... improved QoL was observed [77]. John et al [78] reported ... and self reported pain [78] “
Insert ref number after Wilson et al, Wilson is not 77 but 76, John not 78 but 77, Please correct.
Answer: It has been inserted and corrected.
page 17, line 434 -437: “ total IEQ (100,000) “
What does this mean, only if there are exactly 100,000 IEQ?
Answer: It means total IEQ given in units of 100,000. It has been explained by adding „units” in the manuscript.
page 18, line 437: “Berger et al [89] compared“
Berger is ref 92. Where are refs 89, 90, and 91 in the text?
Answer: They are included in the cited references [83-95] regarding studies on pediatric patients undergoing TPAIT in the worldwide literature. This citation has inserted as follows:
There are numerous studies presenting results of TPAIT in children in the worldwide literature [1,56,57,101-111].
page 18, line 447: “insulin independence (n=14.82%) was“
n=14.82% ? Please correct.
Answer: It has been corrected as follows:
The higher rate of insulin independence (14 of 17 patients, 82%) was noted in children aged ≤8 years compared to independence rate of 41% in 399 patients older than 8 years of age undergoing TPAIT over the same period (p=0.004).
page 18, line 450: “Chinnakotla et al [1], “
Why ref 1, never mentioned before.
Answer: It has been corrected and inserted in the introduction as the first reference.
page 18, line 461: “Gigher insulin independence“
Please correct.
Answer: It has been corrected.
page 19, line 468: “hospitalization (p=0.66). blood loss“
Please correct.
Answer: It has been corrected.
page 19, line 474: “Singh et al [33]“
Wrong name.
Answer: It has been corrected.
page 19, line 476: “Fan et al presented“
Please insert ref number as before.
Answer: It has been inserted.
page 20, line 495: “[1,15,17,64,72, 80, 96-100]“
Ref number 65 is missing.
Answer: It has been inserted as reference 85. The references numbers have been changed due to insertion of additional references.
page 21, line 500: “BMI, body mass index; EIQ, equivalents;“
IEQ not EIQ, equivalents of what?
Answer: It has been corrected as follows: IEQ, islet equivalents..
page 27, line 653: “10.1111/ajt.12354. Epub 2013 Jul 16. PMID: 23859047.Rickels MR, Robertson RP. Pancreatic Islet Transplantation in Humans: Recent“
Rickels MR is a new ref, number? Please correct.
page 28, line 683 and 688: Ref 35 and ref 36 are identical.
Answer: It has been corrected.
page 33, line 814: Ref 59 not mentioned in text.
Answer: It has been inserted (currently reference 79) as follows:
Various complications during and after TPAIT have been reported in the literature. Portal vein thrombosis (PVT) and bleeding are the most important intraoperative complications. As it was mentioned above, a risk of these complications is associated with portal pressure that is associated with islet volume and purity [52,74,79].
page 38 and 39, line 952 - 973: Refs 83, 84,85 and 86 are not mentioned in the text.
Answer: These references (currently 101,102,103,104) regard studies on pediatric patients undergoing TPAIT. They are included in the cited references [previously as 83-95, and currently as 101-111] regarding studies on pediatric patients undergoing TPAIT in the worldwide literature. This citation has inserted as follows: They have been cited and reference numbers have been inserted as follows:
There are numerous studies presenting results of TPAIT in children in the worldwide literature [1,56,57,101-111].

Round 2
Reviewer 1 Report
In this article, the authors reviewed historical aspects, technical aspects, indications as well as outcomes of TPAIT. Their aim is ambitious, since they tried to address several different aspects of this technique.
After the changes made by the authors, this article can be considered for publication.